# Microfluidic characterisation reveals broad range of SARS-CoV-2 antibody affinity in human plasma

Matthias M Schneider[1],*, Marc Emmenegger[2],*, Catherine K Xu[1],*, Itzel Condado Morales[2],*, Georg Meisl[1], Priscilla Turelli[3], Chryssa Zografou[2], Manuela R Zimmermann[1], Beat M Frey[4], Sebastian Fiedler[5], Viola Denninger[5], Raphaël PB Jacquat[1], Lidia Madrigal[2], Alison Ilsley[5], Vasilis Kosmoliaptsis[6,7], Heike Fiegler[5], Didier Trono[3], Tuomas PJ Knowles[1,8], Adriano Aguzzi[2]

The clinical outcome of SARS-CoV-2 infections, which can range from asymptomatic to lethal, is crucially shaped by the concentration of antiviral antibodies and by their affinity to their targets. However, the affinity of polyclonal antibody responses in plasma is difficult to measure. Here we used microfluidic antibody affinity profiling (MAAP) to determine the aggregate affinities and concentrations of anti–SARS-CoV-2 antibodies in plasma samples of 42 seropositive individuals, 19 of which were healthy donors, 20 displayed mild symptoms, and 3 were critically ill. We found that dissociation constants, $K_d$, of anti–receptor-binding domain antibodies spanned 2.5 orders of magnitude from sub-nanomolar to 43 nM. Using MAAP we found that antibodies of seropositive individuals induced the dissociation of pre-formed spike-ACE2 receptor complexes, which indicates that MAAP can be adapted as a complementary receptor competition assay. By comparison with cytopathic effect–based neutralisation assays, we show that MAAP can reliably predict the cellular neutralisation ability of sera, which may be an important consideration when selecting the most effective samples for therapeutic plasmapheresis and tracking the success of vaccinations.

## Introduction

The severe-acute respiratory syndrome coronavirus 2 (SARS-CoV-2) pandemic has not only led to a huge increase in mortality all over the world (1) but has also had a severe impact on health-care systems and socioeconomic indicators. An understanding of the biochemical processes involved in the SARS-CoV-2 infection,

particularly relating to the immune response, is important to best design both treatments and vaccines, as adaptive humoral immune responses are crucial for defending hosts against incoming viruses (2). However, the individual immune responses to any given virus are highly variable and can translate into different efficacies of viral clearance. Several studies have investigated antibodies generated during SARS-CoV-2 infection in the contexts of the immune system (3, 4, 5, 6, 7, 8 Preprint), antibody cross-reactivity (9), disease prevalence in certain geographical areas (10, 11 Preprint, 12 Preprint), and the temporal evolution of the antibody response on the population level (10, 11 Preprint, 12 Preprint). Furthermore, multiple ongoing studies focus on the applicability of antibodies for therapeutic purposes (13), including plasmapheresis (14, 15, 16, 17, 18), which may be a promising therapeutic strategy (18). In such studies, the presence of IgG antibodies is consistently detected within 2 wk after initial infection (4, 5, 11 Preprint).

The biophysical parameters that govern the interaction between any antibody and its cognate antigen are its binding affinity and concentration. Antibody titres are often measured by ELISA of serially diluted samples, yielding a sigmoid dose–response curve, which represents a convolution of antibody affinity and concentration. Samples containing low amounts of high-affinity antibodies can exhibit the same $EC_{50}$ (the dilution yielding half-maximal ELISA signal) as those with large amounts of low-affinity antibodies, yet these two scenarios may result in distinct biological properties.

Surface-plasmon resonance (SPR), in contrast, measures the on- and off-rates of antibodies in the sample binding to the antigen but is unable to decouple antibody concentration and dissociation constants if both are unknown, as in the case in patient serum samples. Although there have been efforts to infer antibody affinities through such approaches (19, 20), these methods are often fraught with large errors, especially when applied in complex samples such as human serum. Moreover, immobilisation-based

[1]Centre for Misfolding Diseases, Yusuf Hamied Department of Chemistry, University of Cambridge, Lensfield Road, Cambridge, UK [2]Institute of Neuropathology, University of Zurich, Zurich, Switzerland [3]School of Life Sciences, École Polytechnique Fédérale de Lausanne, Lausanne, Switzerland [4]Regional Blood Transfusion Service Zurich, Swiss Red Cross, Schlieren, Switzerland [5]Fluidic Analytics, Unit A, Paddocks Business Centre, Cambridge, UK [6]Department of Surgery, Addenbrooke's Hospital, University of Cambridge, Cambridge, UK [7]National Institute for Health Research Blood and Transplant Research Unit in Organ Donation and Transplantation, University of Cambridge, Cambridge, UK [8]Cavendish Laboratory, Department of Physics, University of Cambridge, Cambridge, UK

Correspondence: adriano.aguzzi@usz.ch; tpjk2@cam.ac.uk
*Matthias M Schneider, Marc Emmenegger, Catherine K Xu, and Itzel Condado Morales contributed equally to this work

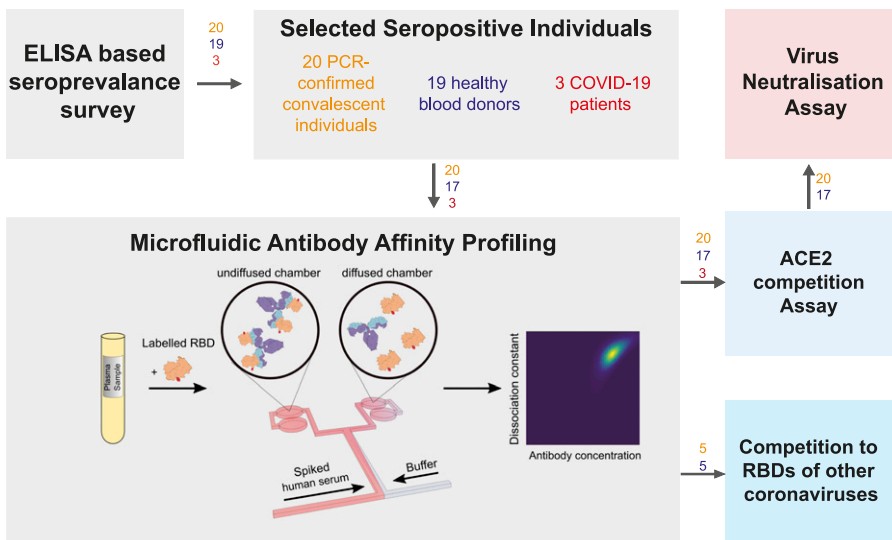

**Figure 1. Principle of the study.**
First, we selected seropositive individuals based on a large-scale seroprevalence survey (11 *Preprint*) and performed four assays: microfluidic antibody affinity profiling, a cytopathic effect–based neutralisation assay, an angiotensin-converting enzyme 2 competition assay and a receptor-binding domain (RBD) cross-reactivity assay. For microfluidic antibody affinity profiling, blood was taken from 42 individuals who underwent an infection with SARS-CoV-2 as confirmed by ELISA. The blood cells were removed by centrifugation and fluorescently labelled RBD protein was added to the plasma, leading to complex formation between the antibodies in the plasma and the extrinsically added fluorescently labelled protein. The average size of fluorescent particles can be inferred from their diffusion rates, providing a readout of the degree of binding. The angiotensin-converting enzyme 2 competition assay and RBD cross-reactivity assay both rely on co-incubation of viral proteins with antibodies and a competitor molecule. The numbers above the arrows represent the number of samples for PCR-confirmed COVID-19–positive individuals (orange), healthy donors who did not undergo PCR testing (blue), and hospitalised COVID-19 patients (red).

techniques such as SPR are prone to surface effects, including surface-aided avidity, the Hook/Prozone effect, and nonspecific binding due to hydrophobic and electrostatic interactions with the surface (21 *Preprint*). As a result, measurement of binding affinities in complex media by surface-based methods is often impossible.

Here, we determined both affinities and concentrations of antibodies to SARS-CoV-2 directly in plasma samples of seropositive individuals using microfluidic antibody affinity profiling (MAAP) (21 *Preprint*). MAAP is a solution-based method which avoids the complications that arise in surface-based techniques. The workflow is represented in Fig 1. We quantified both parameters in 39 seropositive blood donors (initially identified by a high-throughput ELISA technology called TRABI (11 *Preprint*), who presented either mild symptoms or were asymptomatic) and three critically ill, hospitalised patients, demonstrating a comparable antibody response in all 42 patients, independent of the symptoms displayed. In all samples with detectable binding by MAAP, the binding affinity was stronger than the interaction between SARS-CoV-2 spike protein (S) and its associate receptor, the angiotensin converting-enzyme 2 (ACE2), the interaction by which the virus infects host cells (22). Our results are consistent with the hypothesis that the immune response to SARS-CoV-2 is predominantly driven by antibodies that prevent binding of the virus to cellular receptors.

# Results

## MAAP to determine affinities and concentrations in complex solution

We determined the affinities and concentrations of receptor-binding domain (RBD)–reactive antibodies by measuring the equilibrium binding of antibodies in the plasma of seropositive individuals with the RBD directly in solution through MAAP, wherein the effective hydrodynamic radius, $R_h$, of the Alexa 647–labelled RBD protein was monitored. The increase in the measured radius upon complex formation with RBD-reactive antibodies allows the detection and quantification of antibody binding. Such measurements can be performed directly in plasma/serum so that samples are not perturbed by additional purification procedures, and therefore facilitate the quantification and physical characterisation of antibodies, in blood plasma or serum. Here, we used this technique to characterise the immune response of humans to SARS-CoV-2 (21 *Preprint*).

These binding measurements report on the combined polyclonal antibody response and may target different RBD epitopes at different affinities. We therefore simulated the response of a polyclonal sample with two antibodies ("A" and "B") by assuming different concentrations of two antibodies with two different affinities to the RBD (see the Materials and Methods section). We considered three possible scenarios: the concentration of a high-affinity antibody "A" is lower than (Fig 2A), equal to (Fig 2B), or higher than (Fig 2C) that of the weaker binding antibody "B," and model the system both at low and at high RBD concentrations. Only if antibody "A" is present at lower concentration than antibody "B" and RBD is present in excess, "B" contributes predominantly to the observed signal. In all other cases, binding from antibody "A" dominates. Thus, MAAP measurements depend primarily on the concentration of the most affine antibodies, which are also likely to be the most relevant ones to SARS-CoV-2 immunity, assuming that natural antibodies behave analogously to therapeutic, monoclonal antibodies, which are more effective if they have a higher affinity (23).

First, we performed MAAP measurements using CR3022, a well-characterised monoclonal antibody cloned from the lymphocytes of a patient who contracted SARS-CoV (25). The affinity of CR3022 for the SARS-CoV-2 spike protein has previously been determined by SPR to range between 15 and 30 nM in buffer (24). We measured the affinity of CR3022 both in PBS-T and in pooled human plasma of

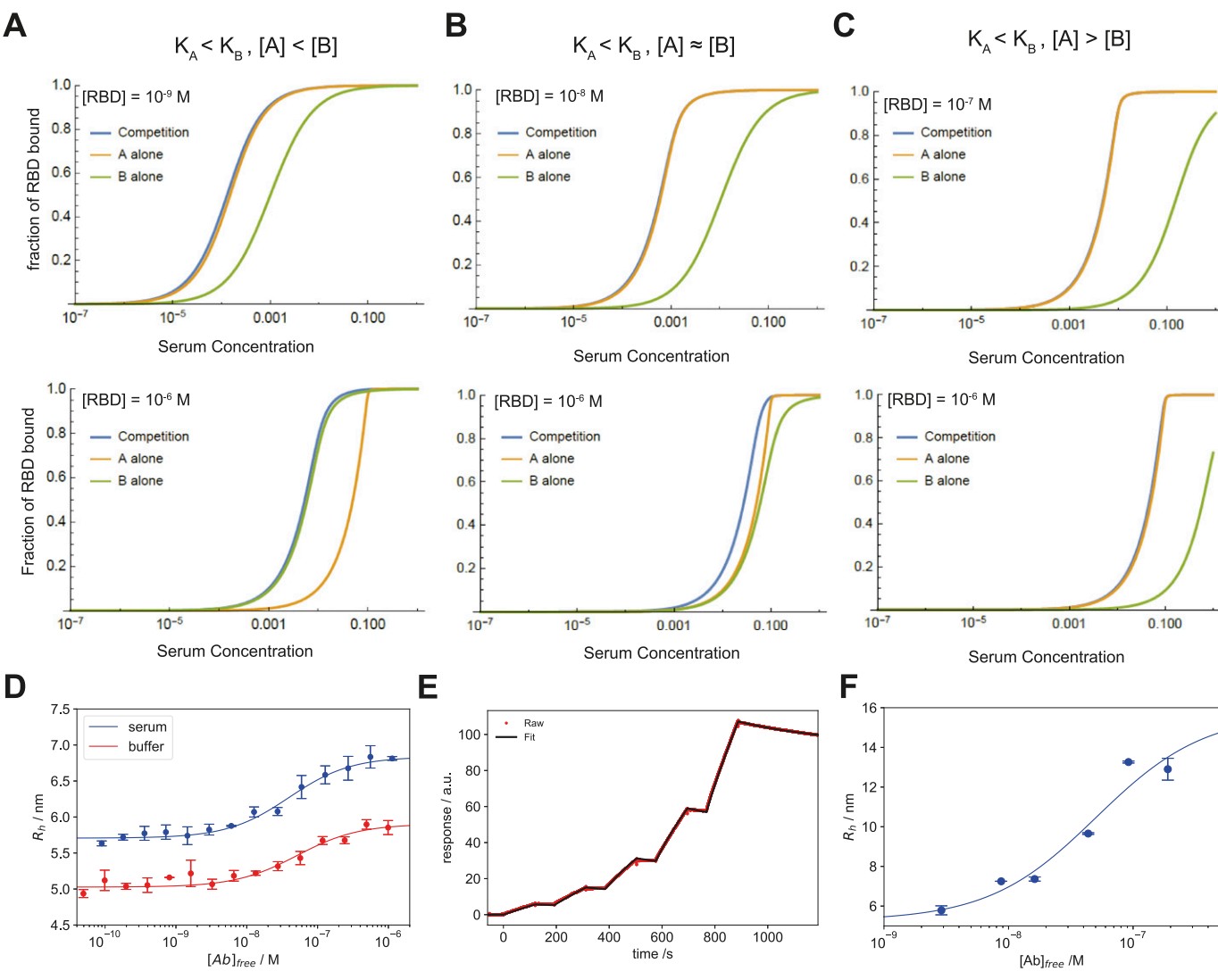

**Figure 2.  Proof of concept.**
**(A, B, C)** Simulation of the competition of two receptor-binding domain (RBD)–reactive antibodies, A and B, with $K_A = 10^{-9}$ M and $K_B = 10^{-7}$ M, respectively. **(A)** For the case in which [A] < [B], there are two sub-regimes: if the RBD concentration is approximately equal to or lower than $K_A$, the combined behaviour resembles that of the stronger binding species A (top). If RBD is present around or above the higher equilibrium constant, $K_B$, then the behaviour resembles that of the weaker binding species (bottom). In between, the behaviour is intermediate. **(A, B)** For the situation where [A] ≈ [B], the combined response is dominated by the tighter binder (A) in both high and low RBD concentration. **(A, C)** For [A] > [B], the signal measured is also determined by the tightly binding antibody (A), regardless of the RBD concentration. **(D)** Binding curve of commercial antibody CR3022 IgG (ab273073, Abcam) in PBS-T (containing 5% HSA [wt/vol]) with RBD yielding a dissociation constant $K_d = 35$ [5, 98] nM, and $K_d = 46$ [10,117] nM in human serum. This is in good agreement with literature values (24). **(E, F)** Binding curve of human-derived anti-SARS-CoV-2 S2 antibody B4 with (E) spike ectodomain using surface-plasmon resonance ($K_d = 1.46 \pm 0.01$ nM) and (F) spike ectodomain with microfluidic antibody affinity profiling, yielding a $K_d = 27$ [12,46] nM. The anti-SARS-CoV-2 S2 domain antibody B4 was labelled with Alexa 647 for the Microfluidic Antibody Affinity Profiling experiment. Data in d and f are represented as mean ± SD of replicate measurements.

healthy subjects by MAAP. The $K_d$ values (Fig 2D) obtained in pure buffer ($K_d = 35$ [5,98] nM, 95% confidence interval given in square brackets) and in serum ($K_d = 46$ [10,118] nM) were both in good agreement with the SPR measurements (24), demonstrating the ability of MAAP to yield consistent measurements both in pure buffer and in a complex matrix, as previously reported for the detection of alloantibodies (21 Preprint).

To further investigate the level of concordance between SPR and MAAP, we additionally cloned an antibody targeting the SARS-CoV-2 Spike S2 domain from a 27-yr-old female blood donor with RT-

PCR–diagnosed COVID-19 with mild symptoms. Memory B cells contained in PBMCs were single-cell sorted by flow cytometry into a 96-well plate using a custom memory B cell panel and were cultured using CD40L$^{low}$ feeder cells. VH, VK, or VL genes of cells from wells tested positive for anti-S2 antibody in ELISA were amplified with nested PCR as previously shown (26). We then expressed the respective sequences as human IgG1 holoantibodies. Among them, the monoclonal antibody termed B4 had a $K_d$ of 1.46 ± 0.01 nM as measured by SPR and of 27 [12,46] nM as measured by MAAP (Fig 2E and F), suggesting broad agreement in the affinity determination

between these two technologies when using monoclonal antibodies.

## Determination of antibody affinity and concentration in plasma of blood donors and patients

As part of a large-scale seroprevalence survey, plasma samples from more than 60,000 cross-departmental hospital patients and healthy donors from the blood donation service (BDS) of the canton of Zurich were investigated for the presence of predominantly IgG antibodies against SARS-CoV-2 S, RBD, and nucleocapsid (NC) proteins (11 *Preprint*) (Fig S1A and B). Seropositivity was defined as having a probability of being seropositive of ≥0.5 using a tripartite immunoassay (11 *Preprint*). To characterise the antibody affinity–concentration relationship, we selected 19 healthy donors with sufficient residual plasma volume with a probability ≥0.85 to be seropositive, who did not undergo screening for SARS-CoV-2 by PCR. In addition, we investigated 20 PCR-confirmed convalescent individuals and three hospitalised patients with acute COVID-19 pneumonia, all of whom were also seropositive (probability ≥0.85) by ELISA (11 *Preprint*). These three patients suffered from diabetes, with patients 2 and 3 presenting additional cardiovascular conditions. The demographic characteristics of the seropositive collective are summarised in Table S1. As an immune target for the downstream MAAP characterisation we selected the RBD of the spike protein because it is crucial for antibody-dependent neutralisation by preventing entry into host cells, and thus may be of significance in the immune response to SARS-CoV-2 (27).

We characterised the serum antibodies from 20 convalescent individuals and 19 seropositive healthy blood donors. Two blood donor samples were excluded due to excessive plasma background fluorescence (I13 and I38) (21 *Preprint*). All convalescent plasma samples and 11 BDS samples displayed an increase in the hydrodynamic radius of the immune complex, indicating binding to RBD (Figs S2A and B and 3A and B). Based on the maximal radius around 6–7 nm reached for most samples (Fig S2), we assume that we have predominantly IgGs with minor contribution of IgAs, whereas IgMs would, based on its molecular weight, display a significantly larger hydrodynamic radius of 8.6 nm or more (28). Six BDS samples did not display any binding to the RBD by MAAP (Fig 3C). Assuming a binding stoichiometry of 1:2 antibody:RBD, we found that 31 MAAP-positive samples displayed antibody concentrations ranging from: 16 to 472 nM, but mostly grouped around tens of nM. In contrast, the $K_d$ values were more variable, ranging from sub-nanomolar (in which case no lower bound $K_d$ could be determined) to 43 nM (Fig 3B). These results are consistent with previous findings on coronaviruses, which showed relatively similar antibody concentrations (29). Furthermore, a previous study by Poulson and co-workers showed that antibody affinities against tetanus toxoid are reported to span several orders of magnitude from the micro-into sub-nanomolar range (30). In addition, there was no significant difference in either $K_d$ or antibody concentration between healthy donors and convalescent patients in our data (Fig S3A–C). For significant binding to occur, the antibody binding site concentration must exceed the $K_d$. Accordingly, our data demonstrate that in all cases where quantifiable binding was detected, the total antibody concentration exceeded the $K_d$ (Fig 3B). As a comparison, we

analysed the three hospitalised patients, which displayed affinities ranging from 2 to 34 nM and antibody concentrations of 4–296 nM; these ranges are similar to those of the non-hospitalised patients (Fig 3B).

Comparing the $K_d$ and concentrations obtained through MAAP with the $pEC_{50}$ values, we observed a weak correlation, indicating that the two methods yield consistent, yet complementary results (Figs 2D and S4). The imperfect correlation is likely to arise from the differences between surface- and solution-based measurements, as discussed previously. Interestingly, there is a good correlation between $pEC_{50}$ with the ratio of antibody concentration to $K_d$ as well as to the antibody concentration itself, whereas it does not show correlation to the $K_d$. This highlights that we are in a strong binding regime, in which ELISA assesses the antibody concentration only (21 *Preprint*).

We next determined the dissociation constant for the interaction between spike protein and ACE2 receptor to be 18 [11, 29] nM (Fig S5A). This is higher than the $K_d$ for most plasma samples of the seropositive individuals, indicating that, in all patients with detectable responses to RBD, the immune response produced antibodies with higher affinity than the virus–receptor interaction.

## Time courses of severely diseased patients

To investigate the importance of mutagenesis and affinity maturation typical in immune response to infection (31, 32, 33), we characterised the antibody affinity and concentration at different stages of the disease in three hospitalised COVID-19 patients, as seen in Fig 3D. Analyses were performed for patient 1 (days post onset of disease manifestation [DPO] 9–13), patient 2 (DPO 8–14), and patient 3 (DPO 7–15). In all cases, no binding was detected until day 12 by MAAP, consistent with the ELISA data (11 *Preprint*) and previous literature (4, 5). Analysis of plasma samples taken from patients 1 and 2, taken 1 and 2 d apart, respectively, indicate that the antibody concentration increases with no change in binding affinity (DPO 12 and 13 for patient 1, and DPO 12, and 14 for patient 2) (Fig 3D). For patient 3, only one time point could be effectively measured (Fig 3D). This finding may imply that on the measured timescale, affinity maturation is not continued beyond these minimal requirements of the antibody–RBD binding being significant enough to out-compete the RBD–ACE2 interaction during the primary immune response and, once this affinity threshold is reached, only the concentration is increased (Fig 3D). This is consistent with previous findings of affinity maturation, which normally occurs after the second exposure to the pathogen (31, 32, 33), and is further supported by the finding that many SARS-CoV-2 antibodies have germline sequences without hyper-affinity maturation in germinal centres (34, 35, 36 *Preprint*).

In conclusion, using the MAAP platform, we were able to investigate the evolution of antibody responses after exposure to SARS-CoV-2; this platform could also be extended to monitor aspects such as the efficacy of vaccines. In-solution measurements avoid the artefacts associated with heterogeneous-phase binding and allow the simultaneous determination of antibody affinities and concentrations under physiologically relevant conditions. The ability of MAAP to independently determine these two fundamental

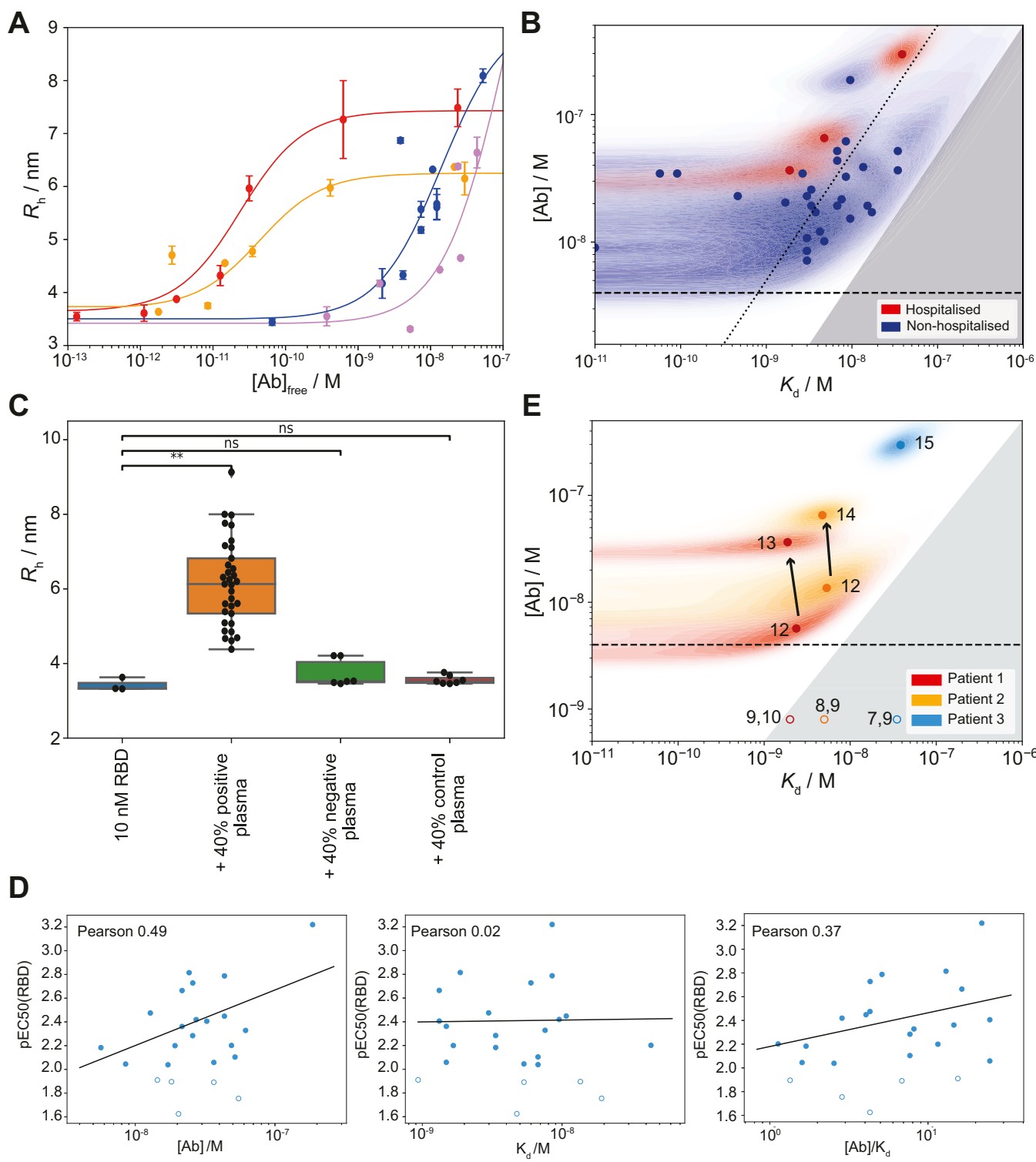

**Figure 3. Affinity and concentration determination in patient plasma.**

**(A, B)** Binding curves for the two samples with the highest and the lowest $K_d$ from panel (B). Tight binders (red curve [$K_d < 4.1 \times 10^{-10}$ M] and yellow curve [$K_d < 6.7 \times 10^{-10}$ M]) are visibly distinguishable from weaker binders (blue curve [$K_d = 8.5 \times 10^{-9}$ M] and purple curve [$K_d = 3.4 \times 10^{-8}$ M]), as they reach the binding transition at lower antibody concentrations. Because a mixture of differently glycosylated antibodies is likely to be present (10), different radii at saturation level are observed for different individuals. The binding curves for all samples are shown in Fig S3. Data are represented as mean ± SD of replicate measurements. **(B)** Probability distributions of dissociation constants, $K_d$, and antibody concentrations, assuming two receptor-binding domain (RBD) binding sites per antibody, for seropositive individuals (blue) and hospitalised COVID-19 patients (red), where significant binding to the RBD was detected. Points correspond to the maximum probability values in the two-dimensional probability distributions (shaded areas). In line with physical principles of binding, binding is not observed for samples with 2[Ab] < $K_d$ (grey region). Notably, some individuals

physicochemical properties of polyclonal antibody responses, thus offers a clear advantage over surface-based techniques.

### Neutralisation and ACE2 receptor–binding competition

SARS-CoV-2 gains access to cells through the ACE2 receptor, and this process can be prevented in cultured cells by antibodies interfering with the binding of the RBD region of the spike protein to ACE2. We therefore investigated the neutralisation capacity of 37 seropositive plasma samples using a wild-type cytopathic effect–based neutralisation assay. Of these, 19 showed neutralisation activity when diluted 1:20, 10 had titres between 1:80 and 1:320, and 4 displayed titres <1:320, whereas four did not show neutralisation even at 1:20 dilution (Figs 4A and S6A and F). Three of these four samples which did not show neutralisation had relatively low titres against RBD by ELISA and also did not show significant binding in the MAAP assay (Tables S2 and S3).

We then compared these results to the inhibitory effect of antibodies directed against the RBD of the spike protein using our microfluidics-based methodology (37 Preprint). We first incubated S1 and fluorescently labelled ACE2; upon complex formation, we incubated this complex with samples of seropositive individuals (Fig 4B and C). The observed hydrodynamic radius of the ACE2 protein, 5.04 ± 0.02 nm, increased during the initial incubation with S1 to 6.25 ± 0.10 nm and remained high when seronegative plasma samples were added, as expected from binding of ACE2 to the S1 protein (Fig 4C). However, this size increase was abrogated by incubation with seropositive plasma samples to 5.06 ± 0.26 nm, indicating that antibodies against RBD can prevent binding of the S1 to ACE2. This was observed for every sample for which we could detect binding by MAAP, with exception of one sample (Fig 4D and E). This suggests that the ACE2 competition assay presented here is a valid and quick tool to determine the neutralisation potential of an antibody and highlights the potential of MAAP for quick, biophysical characterisation of antibodies in solution.

### Cross-reactivity to RBDs of other coronaviruses

To investigate the potential cross-reactivity of SARS-CoV-2 antibodies to RBD from related coronaviruses, we incubated labelled SARS-CoV-2 RBD and unlabelled RBD from other coronaviruses (SARS-CoV, HKU1, and OC43) with seropositive plasma samples. As shown above, the $R_h$ of labelled SARS-CoV-2 RBD increases in the presence of seropositive plasma; we expected competing RBDs to prevent this increase from occurring (see scheme Figs 5A and S5B). We selected 10 randomly chosen plasma samples with measurable

binding by MAAP, including five healthy donors (I10, I11, I16, I18, and I36) and five convalescent individuals (I22, I23, I25, I26, and I28) (Fig 5B). Of these, 24 of 30 combinations showed a decreased radius of more than 10% compared to that in the absence of a competing RBD. In contrast, pre-pandemic plasmas did not show a significant decrease (Fig 5C). Most samples showed strong cross-reactivity (i.e., a large decrease in radius) for at least one of the RBD species. The level of cross-reactivity was strongest towards SARS-CoV RBD in five samples (I10, I18, I22, I28, and I36), towards OC43 RBD in two samples (I23 and I25) and towards HKUI in three samples (I11, I16, and I26). Hence, a potent immune response against one coronavirus may elicit cross-reactive antibodies against RBDs of other coronaviruses. This cross-reactivity could be due to a polyclonal immune response, whereby multiple antibodies against different epitopes on the RBD are produced in the same individual.

These analyses will be useful for studying whether immunity from an infection with one SARS-CoV-2 variant is protective against new variants. Comparisons of differential antibody concentrations and affinities to the variant RBDs may be able to differentiate protective from futile immune responses against SARS-CoV-2 variants and guide the deployment of vaccines and passive immunotherapies.

## Discussion

Antibody responses against a pathogen involve three critical features: The specific epitope that is targeted, antibody concentration, and the affinity of its interaction with the antigen. Although the ratio of the latter two parameters can be determined with a wealth of methods, it is difficult to disentangle affinity and concentration. MAAP allowed us to determine both the concentration and the $K_d$ values of RBD-reactive antibodies in a collective of seropositive subjects whose phenotype ranged from asymptomatic to critically ill. Affinities varied over several orders of magnitude, from subnanomolar to tens of nanomolar. However, in all cases, bar one, where binding in plasma was detectable (i.e., where 2*[Ab] > $K_d$), this interaction was strong enough to prevent the interaction between the ACE2 receptor and the spike protein. This finding was corroborated with a cytopathic effect–based neutralisation assay.

The results detailed above suggest that the MAAP-based competition assay can be used to evaluate passive immunotherapies. For example, antibody affinity is likely to be a key determinant of the efficacy of plasmapheresis. In conclusion, our platform enables the investigation of key biophysical properties of the antibody response to SARS-CoV-2 and other infectious diseases, which in

---

express RBD-reactive antibody such that 2[Ab] ≥ 10$K_d$ (to the right of the dotted line). **(C)** Increase in hydrodynamic radius compared to pure fluorescently labelled RBD (blue) with positive plasma samples (orange), six samples which did not show a size increase (green), and six pre-pandemic control plasma samples (red). Unpaired t test: $P < 0.01$ (**), non-significant (ns). The whiskers show the minimum and maximum values from the distribution. **(D)** Comparison between ELISA (RBD) and microfluidic antibody affinity profiling (MAAP) results for RBD binding, for samples which gave rise both to a peaked probability distribution in both [Ab] and $K_d$ by MAAP, and to a pEC$_{50}$ value greater than two in ELISA. Plots of the pEC$_{50}$ value are shown in comparison to the MAAP-determined ratio of antibody concentration to $K_d$ (left), $K_d$ (middle), and antibody concentration (right). Pearson correlation coefficients are given for each plot. **(E)** Time evolution of $K_d$ and [Ab] probability distributions in patients who required hospitalization; binding was observed by MAAP for three out of four patients investigated. In both patients monitored during the infection (red and orange, filled circles), the antibody concentration increased over time, with no change in binding affinity. Numbered labels indicate the number of days post disease onset (DPO), whereas the grey area represents the region of parameter space in which binding is too low to be measurable by MAAP (2[Ab] < $K_d$). Open circles correspond to earlier time points for which binding was not detectable and position is for illustration purposes only.

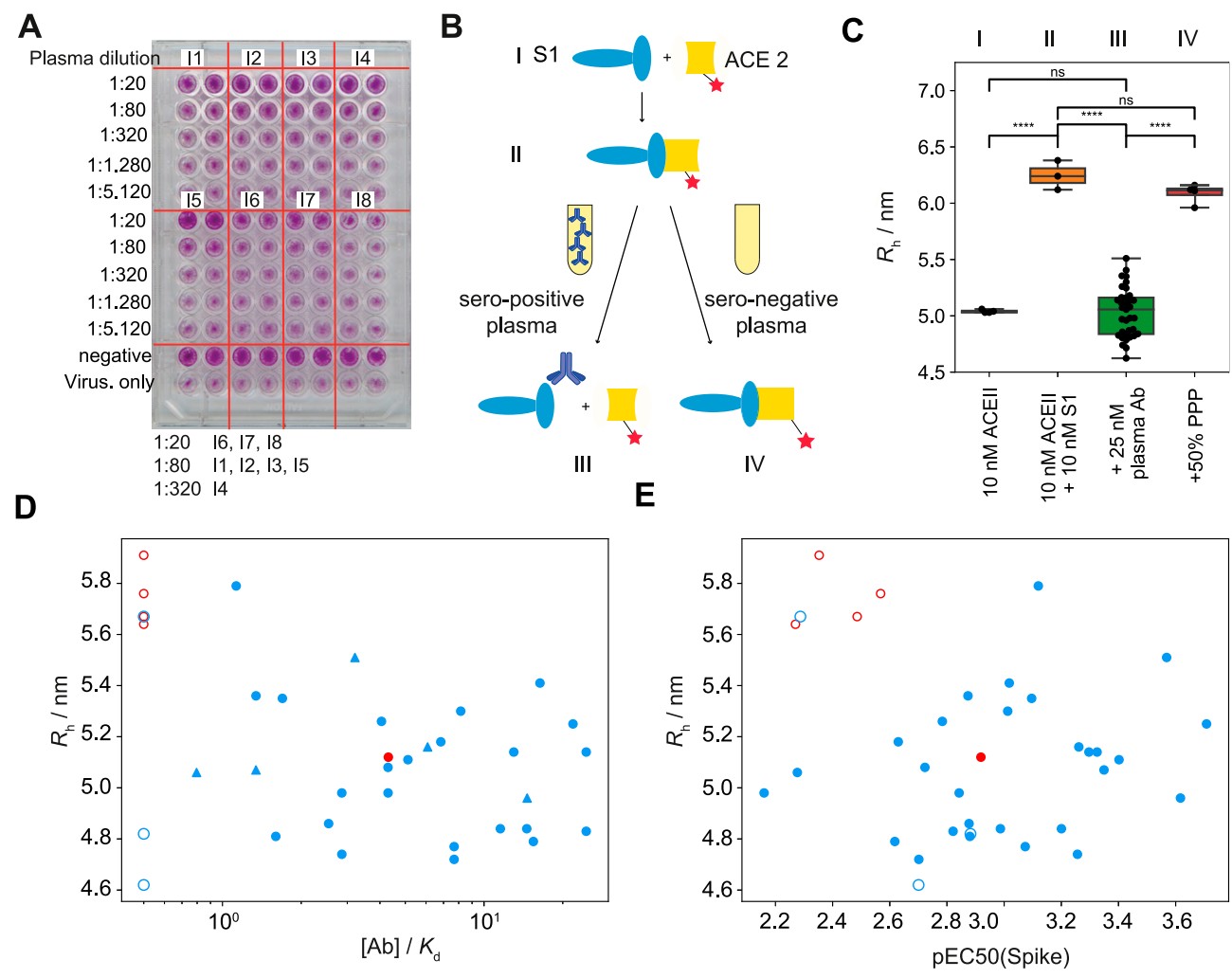

**Figure 4. Angiotensin-converting enzyme 2 (ACE2) competition and cytopathic effect–based neutralisation.**
**(A)** Example plate from a neutralisation assay based on cytopathic effects. We observe neutralisation at a dilution of 1:20 for blood samples from individuals 6, 7, and 8, 1:80 for individuals 1, 2, 3, and 5, and 1:320 for individual 4. All images are shown in Fig S6. **(B)** Schematic of the ACE2 competition assay. We incubated the spike protein with the ACE2 receptor, leading to the formation of the spik–ACE2 complex. Upon the addition of neutralising plasma, this complex is disassembled. **(C)** Hydrodynamic radii of ACE2 in the presence of spike protein in plasma samples of seropositive individuals. When seropositive samples are used, no binding to ACE2 is detected, demonstrating the capability of the antibodies present in plasma to inhibit the interaction relevant for cellular uptake of the virus. By contrast, pre-pandemic plasma samples do not inhibit the spike–ACE2 interaction. Unpaired $t$ test: $P < 0.0001$ (****), non-significant (ns). The whiskers show the minimum and maximum values from the distribution.
**(D, E)** Apparent radius in the ACE2 competition assay compared to the $[Ab]/K_d$ ratio obtained from microfluidic antibody affinity profiling (MAAP) (D) or to ELISA pEC50 (spike) (E) for samples which gave rise to a peaked posterior probability distribution in both $[Ab]$ and $K_d$ (filled circles) and samples for which no binding was observed by MAAP (open circles). The $[Ab]/K_d$ ratio of non-binding samples is assumed to be 0.5, the limit of detection by MAAP, whereas triangles represent the lower bound on $[Ab]/K_d$ for samples which yielded a constrained posterior probability distribution in $[Ab]$, but only an upper bound on $K_d$ by MAAP. Samples which were able to neutralise in the cytopathic effect–based assay are shown in blue, and those incapable of neutralisation at the titres tested are shown in red.

turn may help determining their prognosis and may assist in the development of therapeutic approaches.

# Materials and Methods

### Ethical and biosafety statement

All experiments and analyses involving samples from human donors were conducted with the approval of the local ethics committee (KEK-ZH-Nr. 2015-0561, BASEC-Nr. 2018-01042, and BASEC-Nr. 2020-01731), in accordance with the provisions of the Declaration of Helsinki and the Good Clinical Practice guidelines of the International Conference on Harmonisation.

### Sample collection

EDTA plasma from healthy donors and from convalescent individuals was obtained from the Blutspendedienst (BDS) Kanton Zürich from donors who signed the consent that their samples can be used for conducting research. Samples from patients with COVID-19 were collected at the University Hospital Zurich from patients who signed an informed consent.

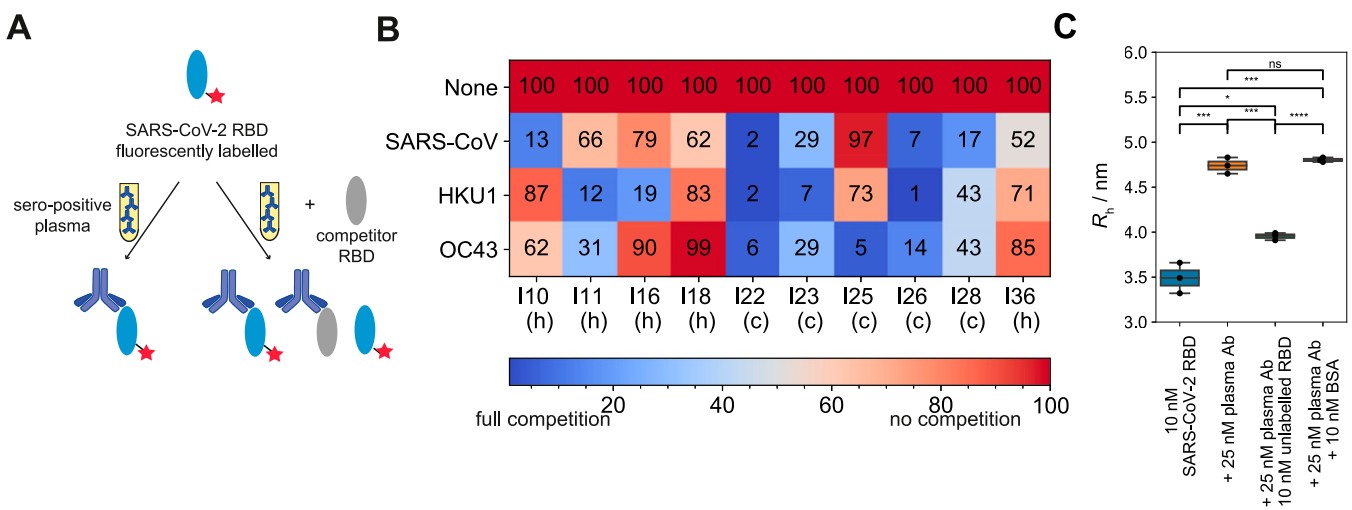

**Figure 5. Cross-reactivity between different receptor-binding domains (RBD)s.**
**(A)** Assay principle. Labelled SARS-CoV-2 RBD was incubated against antibodies from plasma of seropositive individuals. In the absence of any competing RBDs, the binding saturates. In the presence of unlabelled competitor RBD, the antibodies can bind to both the labelled SARS-CoV-2 RBD and the unlabelled competitor RBD, which in turn leads to the presence of unbound labelled SARS-CoV-2 RBD, causing a decrease in the apparent hydrodynamic radius of the mixture of the labelled SARS-CoV-2 RBD. **(B)** Relative decreases in hydrodynamic radii, expressed as percentages, for 10 individuals with different competitor RBDs from SARS-CoV, HKU1, and OC43. 0% indicates that there is no size increase as compared to pure SARS-CoV-2 RBD, meaning that binding of the antibodies to the SARS-CoV-2 RBD is fully inhibited, whereas 100% means that the SARS-CoV-2 RBD-antibody binding was unaffected because there was no competition from the unlabelled RBD. Five samples are from healthy (denoted h) and five from convalescent (denoted c) donors. **(C)** Control experiments for competition assay. 10 nM labelled RBD SARS-CoV-2 was incubated with 25 nM antibodies of plasma samples from seropositive individuals. When incubated in additional presence of 10 nM unlabelled RBD SARS-CoV-2, the radius decreased significantly, whereas the radius remained the same upon addition of BSA. The whiskers show the minimum and maximum values from the distribution. Unpaired $t$ test: $P < 0.0001$ (****), $P < 0.001$ (***), $P < 0.001$ (**), $P < 0.05$ (*), non-significant (ns).

## Reagents

SARS-CoV-2 RBD (SPD-C52H3), ACE2 receptor protein (AC2-H5257), and SARS-CoV-2 spike S1 protein (S1N-C42H4) were purchased from SinoBio and SARS-CoV-2 spike S2 protein from Acro Biosystems. RBDs from SARS-CoV, HKU1, and OC43 were purified as outlined (11 Preprint). CR3022 IgG was purchased from Abcam (ab273073). Microfluidic chips and cartridges for the measurements performed on the Fluidity One-W platform were provided by Fluidic Analytics.

## Labelling

SARS-CoV-2 RBD, S1, B4 mAb, and ACE2 receptor protein were labelled using amine coupling based on NHS chemistry with Alexa Fluor 647 dye. To the protein (typically 1 nmol, 1 equiv.) in 0.1 M NaHCO$_3$ (pH = 8), Alexa Fluor 647 N-hydroxysuccinimide ester (in DMSO, three equiv.) was added. The reaction mixture was incubated for overnight at 4°C, protected from light. The sample was purified by size-exclusion chromatography (Superdex 200 increase) with a flow rate of 0.05 ml/min and PBS as eluent buffer, to yield labelled protein.

## Affinity and concentration determination

MAAP measurements were performed as reported previously. For the MAAP measurements, varying fractions of human plasma samples were added to a solution of the antigen of concentrations varying between 10 and 150 nM, and PBS (containing 0.05% Tween 20, SA) was added to give a constant volume of 20 $\mu$l. The antigen used was RBD-labelled with Alexa Fluor 647 through N-terminal

amine coupling. These samples were incubated at room temperature for 40 min and the size of the formed immunocomplex was determined through measuring the hydrodynamic radius, $R_h$, with microfluidic diffusional sizing using the commercial Fluidity One-W platform. A stream of the fluorescently labelled RBD is introduced onto the microfluidic chip and flowing alongside an auxiliary stream; at low Reynolds number, the two streams mix by diffusion only, so that the fluorescently labelled RBD can diffuse into the auxiliary buffer stream and the two streams are separated at the end of the diffusion channel into two chambers. Because the hydrodynamic radius, $R_h$, is inversely proportional to the diffusion coefficient, $D$, larger proteins or protein complexes show less diffusion into the auxiliary stream than smaller proteins. Taking the fluorescence ratio between diffused and undiffused stream, therefore, allows determination of $R_h$.

The dissociation constant was determined using the following formula: $K_d = [Ab][R]/[AbR]$, where [Ab] and [R] are the equilibrium concentrations of antibody-binding sites and RBD, respectively, and [AbR] is the concentration of bound RBD. The data were analysed by Bayesian inference, according to the following equations. After correction of fluorescence intensities for plasma autofluorescence, the fraction, $f_d$ of RBD to diffuse into the distal channel is defined by reference 38

$$f_d = \frac{[AbR](1-\rho_b) + \left([R]_0 - [AbR](1-\rho_f)\right)}{[R]_0},$$

where $[R]_0$ is the total concentration of RBD and $\rho_b$ and $\rho_f$ are the fractions of bound and free RBD to diffuse into the distal channel,

respectively. By solving the binding equation, we obtain the following expression for $[AbR]$

$$[AbR] = \frac{\alpha[Ab]_{tot} + [R]_0 + K_D - \sqrt{(\alpha[Ab]_{tot} + [R]_0 + K_D)^2 - 4\alpha[Ab]_{tot}[R]_0}}{2},$$

where $\alpha$ is the fraction of plasma used in the measurement and $[Ab]_{tot}$ is the total concentration of antibody binding sites in the sample. $K_d$ and $[Ab]_{tot}$ were thus determined through Bayesian inference, with $\rho_b$ and $\rho_f$ as additional parameters to be inferred. The prior was considered to be flat in logarithmic space for $K_d$ and $[Ab]_{tot}$, flat in linear space for $\rho_b$ and $\rho_f$. The likelihood function was considered to be Gaussian, with an SD obtained through replicate measurements.

To address the question whether either $\log(K_d)$ and/or $\log([AB])$ differ significantly between asymptomatic and convalescent patients, we analysed the likelihoods from Fig S3A with a partially pooled (grouped by symptoms) and fully pooled hierarchical model, following a standard approach as outlined in Gelman et al (39). Effects on $\log([AB])$ and $\log(K_d)$ were analysed separately through marginalisation of the joint probability distributions displayed in Fig S3A. For each parameter, $\theta \in \{\log([AB]), \log(K_d)\}$, we assumed that the effect observed for an individual is the sum of a fixed effect, $\varphi$, and a random effect, $\rho$. For each group (asymptomatic and convalescent in case of partial pooling, all data for the fully pooled model), the fixed effect was assumed to be shared amongst all individuals in that group and the random effects were assumed to be normally distributed across the individuals with zero mean and a shared variance parameter, $\sigma^2$, that is, $\theta = \varphi + \rho$, where $\rho \sim \text{normal}(0, \sigma^2)$. The analysis was performed in a Bayesian framework assuming flat priors for $\varphi$, $\rho$, and $\sigma^2$.

### Simulation of response of polyclonal sample

To simulate the response from a polyclonal sample, we considered a system containing two antibodies and calculated the fraction of RBD bound as a function of the serum concentration. To do so, we assumed both antibodies are in binding equilibrium with RBD:

$$K_A = \frac{[R]_{free}[Ab]_{A,free}}{[Ab]_{A,bound}},$$

$$K_B = \frac{[R]_{free}[Ab]_{B,free}}{[Ab]_{B,bound}},$$

where $K_A$ are $K_A$ are the equilibrium constants, $[Ab]_{A,free}$ and $[Ab]_{B,free}$ are the concentrations of free antibody, $[Ab]_{A,bound}$ and $[Ab]_{B,bound}$ are the concentrations of bound antibody, and $[R]_{free}$ is the concentration of free RBD. Together with the equations for conservation of mass,

$$[R]_{total} = [R]_{free} + [Ab]_{A,bound} + [Ab]_{B,bound},$$

$$[Ab]_{A,total} = [Ab]_{A,free} + [Ab]_{A,bound},$$

$$[Ab]_{B,total} = [Ab]_{B,free} + [Ab]_{B,bound}.$$

This can be solved to give the fraction of bound RBD, $[R]_{bound}/[R]_{total}$, which is plotted in Fig 2 and determines the value measured in MAAP.

### ACE2 competition

S1 protein (10 nM) and ACE2 receptor protein (10 nM) were incubated in PBS for ~40 min. Subsequently, anti-spike antibody in seropositive plasma was added to the mixture to a final antibody concentration of 25 nM and incubated for ~1 h. The hydrodynamic radius was determined by microfluidic diffusional sizing (Fluidity One-W, Fluidic Analytics).

### RBD cross-reactivity competition

Labelled SARS-CoV-2 RBD (10 nM) and was incubated against antibody in a plasma sample, for a final antibody concentration of 25 nM and incubated for ~1 h. Subsequently, an unlabelled competitor RBD was added (10 nM). The hydrodynamic radius was determined by microfluidic diffusional sizing (Fluidity One-W, Fluidic Analytics).

### Affinity determination of monoclonal antibodies by MAAP

10 nM of Alexa Fluor 647 labelled SARS-CoV-2 spike S1 (S1N-C52H4, ACROBiosystems) was mixed with increasing concentrations of CR3022 IgG in the presence of 90% heat-inactivated human serum (H5667; Merck) and in pure PBS. The interactions between ECD Spike and the Alexa Fluor 647–labelled SARS-CoV-2 S2 monoclonal antibody B4 were monitored using 10 nM of the labelled B4 antibody and varying concentrations of ECD Spike, from 0 up to 100 nM to reach saturation. The labelled species was combined with unlabelled antigen, and incubated for 20 min before performing the diffusion measurements at 25°C in PBS (pH 7.8 with 0.05% Tween 20).

### Cytopathic effect–based neutralisation assay

The day before infection, VeroE6 cells were seeded in 96-well plates at a density of 12,500 cells per well. Heat-inactivated plasma samples from seropositive individuals were diluted 1:20 in DMEM 2% FCS in a separate 96-well plate. Fourfold dilutions were then prepared until 1:5,120 in DMEM 2% FCS in a final volume of 60 $\mu$l. SARS-CoV-2 viral stock ($2.4 \times 10^{-6}$ PFU/ml) diluted 1:100 in DMEM 2% FCS was added to the diluted sera at a 1:1 volume/volume ratio. The virus-plasma mixture was incubated at 37°C for 1 h. Then 100 $\mu$l of the mixture was subsequently added to the VeroE6 cells in duplicates. After 48 h of incubation at 37°C cells were washed once with PBS and fixed with 4% fresh formaldehyde solution for 30 min at RT. Cells were washed once with PBS and plates were put at 58°C for 30 min before staining with 50 $\mu$l of 0.1% crystal violet solution for 20 min at RT. Wells were washed twice with water and plates were dried for scanning. A negative pool of sera from pre-pandemic healthy donors was used as negative control. Wells with virus only were used as positive controls.

### Surface plasmon resonance

The interaction of SARS-CoV-2 S2 monoclonal antibody B4 to the spike ectodomain (see reference 11 *Preprint*) was measured on a Biacore T200. Serially diluted (16–1 nM) monoclonal antibody B4 was injected at a flow rate of 50 µl/min for association, and disassociation was performed over a 600-s interval. The affinity was calculated using a 1:1 Langmuir binding fit model.

### Flow cytometry

Cryopreserved PBMCs were thawed, washed, and resuspended in IMDM medium containing 10% FCS and antibiotics. Total B cells were enriched using negative selection with immunomagnetic beads according to the manufacturer's instructions (STEMCELL Technologies). Cells were stained with fluorescently labelled antibodies against CD3 (APC-Cy7;HIT3a), CD14 (APC-Cy7;M5E2), CD19 (PE-Cy7;SJ25C1), IgD (FITC;IA6-2), CD27 (PE;M-T271), CD38 (V450;HB7) (all BioLegend), and memory B cells were sorted with a BD FACS Melody Cell Sorter in a Biosafety Level III facility.

### In vivo B cell cultures

Single memory B cells were sorted and cultured in IMDM medium containing 10% 55 µM HI-FBS, 50 µM 2-mercaptoethanol, 2 mM L-glutamine, 100 U/ml penicillin, 100 mg/ml streptomycin, 1 mM sodium pyruvate, and 1% MEM nonessential amino acids (all Invitrogen). The medium was supplemented with 10 ng/ml of IL-2, IL-21, and IL-6 (all Peprotech) in multiple round-bottom 96-well culture plates pre-seeded with CD154-expressing stromal cells (CD40L-low cell line, kind gift from Dr Xin Luo).

### Cloning and expression of immunoglobulin genes

Cells from wells tested positive for anti-S antibody in ELISA assays were collected and subjected to RNA extraction (RNeasy mini kit, QIAGEN). cDNA synthesis was performed as previously described and VH, VK, or VL genes were amplified in two PCR reactions with primer mixes designed to amplify the different heavy- and light chain families (26). For antibody expression, HEK293A was transfected with plasmids carrying the human constant regions (IgG1) and cloned with the variable heavy and light chain sequences. The supernatant was harvested and Protein G Sepharose 4 Fast Flow beads (GE Healthcare Life Sciences) were used for antibody purification as described (40).

## Data Availability

The raw data underlying this study will be made available upon reasonable request. The biobank samples are limited and were exhausted in several instances. Therefore, while we will make efforts to provide microliter amounts of samples to other researchers, their availability is physically limited.

## Supplementary Information

## Acknowledgements

All authors wish to thank their entire teams for support in the laboratory. We thank Tom Scheidt (IMB, Mainz) who provided graphics for Fig 1. We are grateful to Aaron Ring, John D Huck, and Feimei Liu (Yale School for Medicine) for sharing the SARS-CoV, HKU1, and OC43 RBD proteins. Blood of COVID-19 patients from the USZ was acquired with the help of Irina L Dubach and Dominik I Schaer, whom we kindly acknowledge. We thank Laryssa Kovtonyuk (Hematology, USZ) for help with flow cytometrical sorting, Jens Sobek (FGCZ, Zurich) for SPR measurement, Eméry Schindler (BDS, Zurich) for information on the blood donors, Lidia Madrigal (Neuropathology, USZ) for MAAP measurements, and Panos Stathopoulos and Kevin C O'Connor (Yale University Department of Immunobiology) for advice on antibody cloning. We are grateful to all blood donors and hospital patients for helping us conduct this study. We acknowledge financial support from the Biotechnology and Biological Sciences Research Council to TPJ Knowles, as well as the Frances and Augustus Newman Foundation to TPJ Knowles; the ERC PhyProt (agreement no. 337969) to MM Schneider, CK Xu, MR Zimmermann, G Meisl, and TPJ Knowles; the Centre for Misfolding Diseases, Cambridge to MM Schneider, CK Xu, G Meisl, and TPJ Knowles; St John's College Cambridge to MM Schneider, MR Zimmermann, and TPJ Knowles; as well as CK Xu and MR Zimmermann from the Herchel Smith Fund; I Condado Morales acknowledges funding from the Swiss FCS and the Forschungskredit of the University of Zurich. Institutional core funding by the University of Zurich and the University Hospital of Zurich, as well as Driver Grant 2017DRI17 of the Swiss Personalized Health Network to A Aguzzi; funding by grants of Innovation Fund of the University Hospital Zurich (INOV00096), and of the NOMIS Foundation, the Schwyzer Winiker Stiftung, and the Baugarten Stiftung (coordinated by the USZ Foundation, USZF27101) to A Aguzzi and M Emmenegger. V Kosmoliaptsis acknowledges funding from NIHR (PDF-2016-09-065) and from a PI Terasaki Scholarship.

### Author Contributions

MM Schneider: conceptualization, data curation, formal analysis, supervision, investigation, visualization, methodology, project administration, and writing—original draft, review, and editing.
M Emmenegger: conceptualization, data curation, formal analysis, supervision, investigation, visualization, methodology, project administration, and writing—original draft, review, and editing.
CK Xu: conceptualization, software, formal analysis, investigation, visualization, methodology, project administration, and writing—original draft, review, and editing.
I Condado-Morales: data curation, formal analysis, investigation, methodology, project administration, and writing—original draft, review, and editing.
G Meisl: conceptualization, software, formal analysis, supervision, methodology, and writing—original draft, review, and editing.
P Turelli: data curation, formal analysis, visualization, methodology, and writing—review and editing.
C Zografou: data curation, formal analysis, methodology, and writing—review and editing.
MR Zimmermann: formal analysis and writing—review and editing.
BM Frey: conceptualization, resources, methodology, project administration, and writing—review and editing.

S Fiedler: resources, data curation, methodology, and writing—review and editing.

V Denninger: resources, data curation, methodology, and writing—review and editing.

RPB Jacquat: formal analysis and writing—review and editing.

L Madrigal: data curation and writing—review and editing.

A Ilsley: data curation and writing—review and editing.

V Kosmoliaptsis: resources, methodology, and writing—review and editing.

H Fiegler: conceptualization, resources, methodology, and writing—review and editing.

D Trono: conceptualization, resources, supervision, methodology, and writing—review and editing.

TPJ Knowles: conceptualization, resources, supervision, funding acquisition, methodology, and writing—original draft, review, and editing.

A Aguzzi: conceptualization, resources, supervision, funding acquisition, project administration, and writing—original draft, review, and editing.

## Conflict of Interest Statement

TPJ Knowles is a member of the board of directors of Fluidic Analytics. A Aguzzi is a member of the board of directors of Mabylon AG which has funded antibody-related work in the Aguzzi lab in the past. V Denninger, S Fiedler, H Fiegler are employees of Fluidic Analytics, MM Schneider, CK Xu, G Meisl and V Kosmoliaptsis are consultants. All other authors declare no competing interests.

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
