## [Reviewer comments · Life Science Alliance]

Life Science Alliance

Microfluidic Characterisation reveals Broad Range of SARS-CoV-2 Antibody Affinity in Human Plasma

Matthias Schneider, Marc Emmenegger, Catherine Xu, Itzel Condado-Morales, Georg Meisl, Priscilla Turelli, Chryssa Zografou, Manuela Zimmermann, Beat Frey, Sebastian Fiedler, Viola Denninger, Raphael Jacquat, Lidia Madrigal, Alison Ilsley, Vasilis Kosmoliaptsis, Heike Fiegler, Didier Trono, Tuomas Knowles, and Adriano Aguzzi

DOI: <https://doi.org/10.26508/lsa.202101270>

Corresponding author(s): Adriano Aguzzi, University of Zurich

Review Timeline:

Submission Date:	2021-10-22
Editorial Decision:	2021-11-10
Revision Received:	2021-11-13
Accepted:	2021-11-15

Scientific Editor: Novella Guidi

Transaction Report:

Please note that the manuscript was reviewed at *Review Commons* and these reports were taken into account in the decision-making process at *Life Science Alliance*.

Review
COMMONS

November 10, 2021

RE: Life Science Alliance Manuscript #LSA-2021-01270

Prof. Adriano Aguzzi
University Hospital Zurich
Dept. of Pathology
Institute of Neuropathology
Zurich, ZH 8091
Switzerland

Dear Dr. Aguzzi,

Thank you for submitting your revised manuscript entitled "Microfluidic Affinity Profiling reveals a Broad Range of Target Affinities for Anti-SARS-CoV-2 Antibodies in Plasma of Covid Survivors". We would be happy to publish your paper in Life Science Alliance pending final revisions necessary to meet our formatting guidelines.

- please upload your main and supplementary figures as single files;
- please add a Running Title and a Summary Blurb/Alternate Abstract in our system
- please add a Category for your manuscript in our system
- please add the twitter handle of your host institute/organization as well as your own or/and one of the authors in our system
- please make sure the list of the authors and their order in your manuscript and our system match
- please add your main, supplementary figure, and table legends to the main manuscript text after the references section
- please consult our manuscript preparation guidelines <https://www.life-science-alliance.org/manuscript-prep> and make sure your manuscript sections are in the correct order
- please separate the Results and Discussion section into two - 1. Results 2. Discussion, as per our formatting requirements
- please use capital letters when introducing panels in figures, their legends, and callouts in the manuscript text
- please add Author Contributions for your manuscript in our system, as well
- please add a conflict of interest statement to your main manuscript text
- we encourage you to revise the figure legend for figure S6 such that the figure panels are introduced in an alphabetical order
- please upload your Tables in editable .doc or excel format
- please add callouts for Figures 5C, S1A-B, S2A-B, S3B-C, S5A-B, S6A-F, S7A-C to your main manuscript text

A. FINAL FILES:

-- Summary blurb (enter in submission system): A short text summarizing in a single sentence the study (max. 200 characters including spaces). This text is used in conjunction with the titles of papers, hence should be informative and complementary to the title. It should describe the context and significance of the findings for a general readership; it should be written in the

present tense and refer to the work in the third person. Author names should not be mentioned.

B. MANUSCRIPT ORGANIZATION AND FORMATTING:

Sincerely,

Reviewer #1 (Comments to the Authors (Required)):

This article focuses on the utilization of a microfluidics-based approach to assess the interaction affinity kinetics of complex antibody mixtures from patient samples to the RBD region of the SARS-CoV-2 Spike protein. The authors find that their platform is capable of reasonably estimating dissociation constants for the overall antibody mixture in a given plasma sample. Furthermore, they provide evidence that while antibody concentrations in plasma were similar across patients, the Kd of the plasma for RBD varied considerably. Using this approach, they show that in a modestly sized cohort, the aggregate antibody affinity of a plasma sample must be tighter than the S1/ACE2 interaction affinity to enable viral neutralization function.

The authors have diligently answered all my prior concerns. I have no further comments or requests. I do not feel any further issues need to be addressed.

November 15, 2021

RE: Life Science Alliance Manuscript #LSA-2021-01270R

Prof. Adriano Aguzzi
University of Zurich
Dept. of Pathology
Institute of Neuropathology
Zurich, ZH 8091
Switzerland

Dear Dr. Aguzzi,

Thank you for submitting your Research Article entitled "Microfluidic Characterisation reveals Broad Range of SARS-CoV-2 Antibody Affinity in Human Plasma". It is a pleasure to let you know that your manuscript is now accepted for publication in Life Science Alliance. Congratulations on this interesting work.

DISTRIBUTION OF MATERIALS:

Again, congratulations on a very nice paper. I hope you found the review process to be constructive and are pleased with how the manuscript was handled editorially. We look forward to future exciting submissions from your lab.

Sincerely,
